# Design and evaluation of low-cost, DIY programmable tissue processor for solvent exchange in biological sample preparation

May Wang[1], Samantha Pelletier[2], Alexis Ellis[3], Robert F. Shepherd[1], Margaret H. Frank[2], Abraham D. Stroock[2,3,4], Anand Kumar Mishra[1,5]*, Vesna Bacheva[2,3,4]*

**1** Department of Mechanical and Aerospace Engineering, Cornell University, Ithaca, New York, United States of America, **2** School of Integrative Plant Science, Cornell University, Ithaca, New York, United States of America, **3** Smith School of Chemical and Biomolecular Engineering, Cornell University, Ithaca, New York, United States of America, **4** Kavli Institute at Cornell for Nanoscale Science, Cornell University, Ithaca, New York, United States of America, **5** Department of Mechanical Materials and Aerospace Engineering, West Virginia University, Morgantown, West Virginia, United States of America

☯ These authors contributed equally to this work
* am2877@cornell.edu (AM); vb333@cornell.edu (VB)

## Abstract

Imaging techniques are fundamental tools in biology for examining cell growth and responses to the environment. Many tissues require fixing, staining, and/or clearing before they can be visualized under a microscope. However, these protocols, such as those using propidium iodide (PI), a fluorescent cationic stain widely used across biological specimens including plant, mammalian, and bacterial, often require laborious dehydration and rehydration steps to facilitate stain penetration. These stepwise solvent exchanges, for example, by passing tissues through a graded ethanol series, are time-consuming and manually intensive. While automated tissue processors offer an alternative, they are outside of the budget for many labs. Here, we present an open-source, low-cost (~$400) automated tissue processor that performs sequential dehydration and rehydration of biological tissues, significantly reducing hands-on labor. The processor is made of readily available, standardized parts and includes custom software that allows users to define and save protocols. We demonstrate the use of the processor by automating a multi-day PI staining protocol across multiple plant species, tissue morphologies, and users, and by comparing tissue quality with hand-processed samples. Our design provides a low-cost, accessible alternative to expensive commercial tissue processors, offering a practical solution for a wide range of biology laboratories.

## Introduction

Most biological sciences heavily rely on imaging techniques that are fundamental for understanding cell growth, differentiation, cell death, and response to environmental

**Data availability statement:** All relevant data are within the paper and its Supporting Information files.

**Funding:** This work was supported by the National Science Foundation STC Center for Research on Programmable Plant Systems (grant DBI-2019674). V.B. was supported by Schmidt Science Fellows, the Swiss National Science Foundation Postdoc.Mobility fellowship (grant 214477), and the KIC Postdoctoral Fellowship. There was no additional external funding received for this study. The funders had no role in study design, data collection and analysis, decision to publish, or preparation of the manuscript.

**Competing interests:** The authors have declared that no competing interests exist.

cues [1]. Biological tissues are often inherently transparent or lack contrast, and introducing stains or dyes is essential to visualize their cellular structures [2–4]. For example, propidium iodide (PI) is a fluorescent cationic stain that is membrane-impermeant and selectively enters cells with compromised membranes (i.e., dead or damaged cells). In animal cells, it binds to DNA and RNA in the nucleus, while in plant cells, it binds to negatively charged pectin in the cell wall, emitting red fluorescence in both cases [5]. Such stains greatly enhance image contrast, but their application typically requires elaborate sample preparation.

Many staining protocols involve sequential dehydration and rehydration steps to incrementally replace the tissue's fluid environment, ensuring thorough infiltration of dyes into the tissue, and enabling subsequent tissue clearing [6]. For instance, a PI staining protocol for plant non-transparent tissues involves moving samples through a graded series of ethanol solutions (from water up to 100% ethanol and back down), with each step lasting on the order of 20–30 minutes. In practice, implementing such a protocol requires manually transferring solvent over several hours or days [7,8]. This manual process is not only labor-intensive and tedious but also introduces risk of variability (e.g., tissues drying out if a step is delayed or the solvent concentration is miscalculated).

Automated tissue processors are commercially available to handle sequential solvent exchanges in histology and pathology labs, and in principle could execute these staining sequences. However, existing commercial tissue processors are often expensive and engineered for far more complex tasks than needed for routine biology workflows [9]. High-end histology processors, which automate dehydration, clearing, and paraffin infiltration, can cost on the order of tens of thousands of US dollars [10]. These processors typically include additional features like vacuum and heating for embedding tissues in wax, and can process large batches of samples overnight. While powerful, such systems are financially out of reach for many research groups and are overbuilt for protocols that only require solvent exchanges. In recent years, many biological workflows have been automated at much lower cost using 3D-printed parts and off-the-shelf components [11–16]. Specifically, open-source liquid-handling platforms and microfluidic systems have enabled automation of pipetting and microscale fluid control [17,18]. However, these approaches are not designed for centimeter-scale tissues, and to our knowledge no low-cost system exists for automated solvent exchange at this scale.

We here developed a low-cost, open-source tissue processor that automates sequential solvent exchanges for biological tissue (Fig 1). The processor can be constructed for approximately $400 in parts and materials, using off-the-shelf components commonly found in makerspaces and a few custom 3D-printed pieces. We provide detailed instructions and design files so that other researchers can easily replicate and assemble the processor. The processor uses a microcontroller-based control system to automate solvent mixing and transfer, thus eliminating the need for constant human intervention. The custom software allows users to define protocols (series of steps with specified solvent concentrations and durations). While we demonstrate the processor's capabilities on a propidium iodide staining protocol in

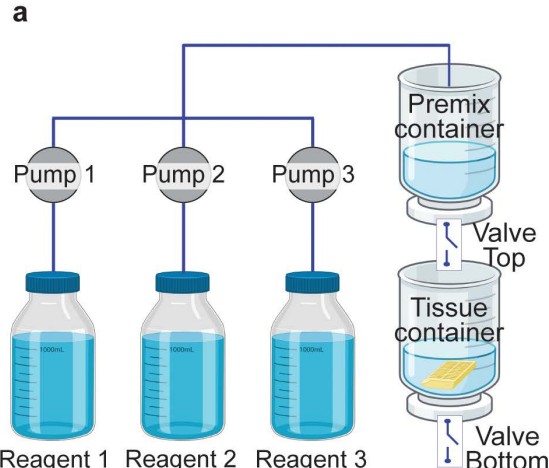
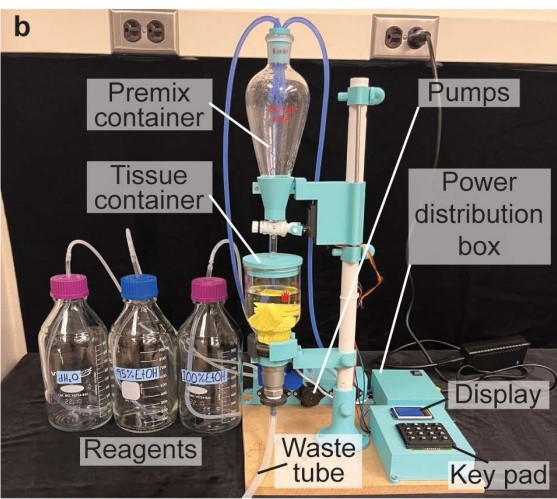

**Fig 1. Overview of system. (a)** Schematic diagram and **(b)** photographs of the automated tissue processor. The system consists of two main containers: a premix container for preparing solvents and a tissue container for processing. Using three peristaltic pumps, solvents are mixed to the desired concentration in the premix container and transferred to the tissue container via an automated top ball valve. After the tissue soaks for a programmed time, the solution is drained into a waste bottle through a bottom valve, and the next solution is introduced. The process is fully programmable via a keypad and display.

this work, the system is versatile and can be programmed for different protocol requiring a series of solvent exchange steps (for example, other histological stains or tissue clearing procedures).

## Materials and methods

### Bill of materials

Table 1 lists all off-the-shelf components with typical vendor sources and prices for each component. Where possible, we opted for widely used modules (for instance, an Arduino Mega microcontroller and Adafruit peristaltic pumps) to maximize compatibility and community support.

### Fabrication

Most components were fabricated with a fused-deposition modeling (FDM) printer, the Bambu Labs X1C, and a digital-light-processing (DLP) printer, the Carbon M1, chosen for their fabrication flexibility and high-resolution capabilities. We could also use more affordable SLS printers, such as an Elegoo printer, for high-resolution component printing. Additionally, standard machining methods could be used to fabricate the same components. Table 2 lists every printed component along with the recommended printer, material, and any notable print settings. The combined print time for all parts was about fifteen to twenty hours on a hobby-grade machine and can be reproduced in a typical makerspace or fabrication facility. We used polylactic acid (PLA) with the FDM printer and UMA-90 urethane resin with the Carbon printer to produce the parts. The CAD models were developed using SolidWorks 2022 (SolidWorks Corp.) and are provided in S1 File in both STEP and SLDPRT formats to facilitate modification and reuse. Due to its complexity, a separate render of the glassware tower has been provided in Fig 2, along with specific printing instructions in Table 2.

### Assembly procedure

The assembly can be conceptually divided into four subsystems: (1) the glassware tower (shown in Fig 2) holding the premix and tissue container, (2) the control box containing the user interface (display and keypad), (3) the relay and power

**Table 1. List of off-the-shelf components including electrical and mechanical components.**

| Components | Price $ | Link | Components | Price | Link |
|---|---|---|---|---|---|
| **Electrical Components** | | | **Mechanical Components** | | |
| Keypad | 5.95 | Adafruit | Graduated Filtration Funnel | 104.5 | Amazon |
| Display | 24.95 | Adafruit | Additional Funnel | 21.97 | Amazon |
| Arduino Mega | 48.9 | Amazon | Steel Rod | 6.99 | Amazon |
| MG995 Positional Servo | 19.6 | Amazon | 0.75" ID, 0.85" OD PVC Pipe | 10.96 | Home Depot |
| Peristaltic Pump | 24.95 | Adafruit | 12"x12"x1/2" MDF Board | 6.97 | McMaster |
| Relay Board | 9.99 | Amazon | 3.5mm ID Silicone tubing | 7.99 | Amazon |
| 9V 5A Power Adapter | 7.49 | Amazon | 7.5 mm ID Silicone Tubing | 11.99 | Amazon |
| Buck Converter | 7.99 | Amazon | Teflon Tape | 2.99 | Amazon |
| Motorized Ball Valve | 35.9 | Amazon | 225 Oring | 5.91 | Amazon |
| Wago Connectors | 19.99 | Amazon | Wood Screws | 6.99 | Amazon |
| Barrel Jack Connector | 0.95 | Adafruit | Metric Screws | 15.59 | Amazon |
| 22 AWG Wire | 12.99 | Amazon | | | |
| **Total cost** | | | **422 $** | | |

**Table 2. 3D printing details of glassware tower components with manufacturing notes.**

| Glassware Column | | | | |
|---|---|---|---|---|
| **Number** | **Name:** | **Qty:** | **Material** | **Note** |
| 1 | Top Silicone Receiver | 1 | PETG | |
| 2 | Servo Holder | 1 | PETG | |
| 3 | Separatory Funnel Holder | 1 | PETG | May need to adjust the angle of cone to fit specific glassware. Tap M3 screw directly into plastic.<br>Print left and right part by separating into objects |
| 4 | NPT Lid | 1 | PETG | Installation 225 o-ring. |
| 5 | NPT Angle Locker | 3 | PETG | PLA may be used to have more strain. Tradeoffs between sealing force and rigidity may be made. |
| 6 | Chamber NPT Adapter | 1 | Resin | Cut a 225 o-ring, trim to length and press into face seal groove. |
| 7 | NPT Arm | 1 | PETG | |
| 8 | NPT Holder | 1 | PETG | Tap M3 screw directly into plastic.<br>Print left and right part by separating into objects |
| 9 | NPT Hose Barb | 1 | Resin | Careful with the hose barb, relatively fragile. |
| 10 | PVC Base | 1 | PETG | Use wood screws to secure to base plate. |
| 11 | Assembly Pin | 5 | PETG | Print pins horizontally to prevent shearing. |
| 12 | Rail Holder | 2 | PETG | Connect with steel rod. |
| 13 | Servo Connector | 2 | PETG | (Not Pictured) These slide on to the ends of the separatory funnel's twisting mechanism, and are screwed into the servo's rotating arms. |

box housing the electronics and power distribution, and (4) the pump module securing the peristaltic pumps and associated tubing. We assembled each subsystem separately and then integrated them onto the base platform.

For the glassware tower, a 0.75″ inner-diameter PVC pipe serves as the vertical column to which we drilled 13/32″ holes to mount the 3D-printed holders of the tissue and premix containers. We sealed all glass joints between the containers and valves with Teflon tape and O-rings to prevent leaks. During assembly, we verified that the motorized ball valve at the top of the tissue container and the servo-actuated drain at the bottom both formed tight seals.

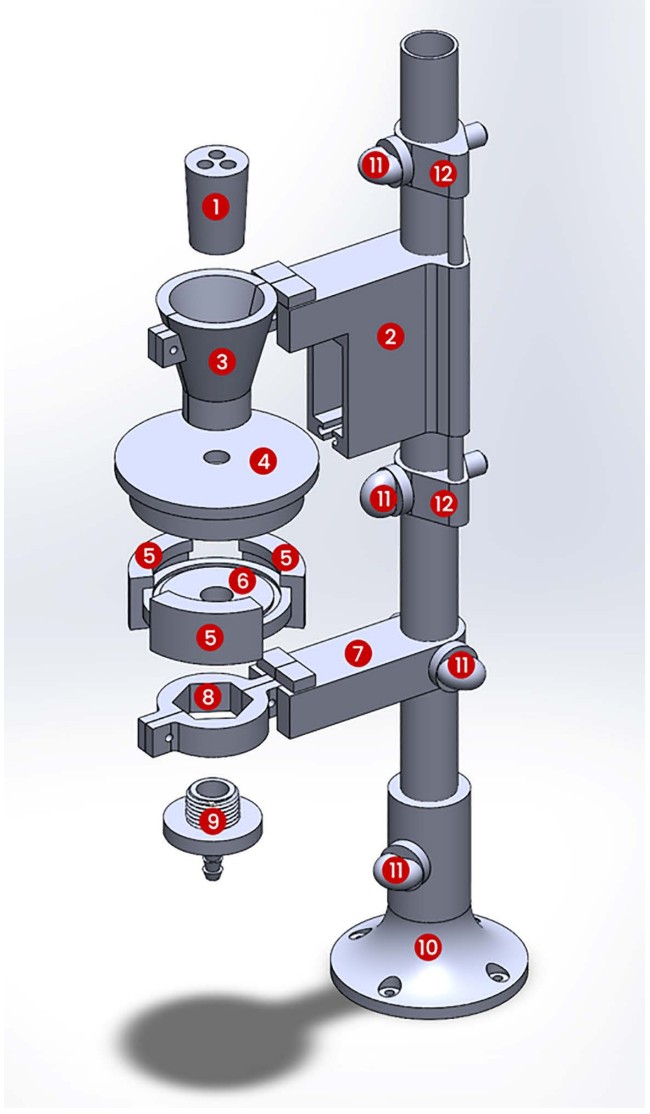

**Fig 2. 3D CAD rendering of the tower assembly showing the mounting structure for the glass containers that serve for premixing.** The labels correspond to part numbers listed in Table, which provides 3D printing details and fabrication notes.

In the control box, we mounted an LCD screen and a 4x4 matrix keypad that serve as the user interface. These components snap into a 3D-printed panel, and we secured them with machine screws. To avoid damaging the plastic threads in the printed material, we pre-threaded the holes by gently driving the screws in once before attaching the electronics. The Arduino Mega board sits behind the display and connects to it and the keypad via jumper wires. We used header pins and ribbon cables instead of soldering directly to the display/keypad, in order to make replacements or reconfigurations easier. All the wiring from the control box to other parts of the system is bundled and routed through a side opening.

The relay and power box contains the power input, switching relays, and voltage regulators. This box is the most densely packed. We selected a compact 4-channel relay board to drive the pumps and the valve actuators, which draw higher currents than the microcontroller can supply directly. The Arduino controls these relays to turn the pumps on/off and to open/close the valves. For power, a single 9V DC, 5A adapter feeds the system; inside the box, two buck converter

modules step this down to 5V for the servo and microcontroller and to 6V for the peristaltic pumps, which ran optimally at ~6V in our tests. We added an inline switch on the 9V input for convenience, mounted on the side of the box. The wiring in this section was organized using small lever-style connectors (Wago connectors) to join common lines and to allow easy disconnection of components. Zip ties helped bundle excess wire and keep the layout neat to avoid any shorts. A schematic of all electrical connections is shown in Fig 3, and the complete pin assignments of the Arduino to each component are listed in Table 3 for reference. In brief, the microcontroller uses separate digital output pins to toggle each of the three pumps, the top ball valve, which is an electrically controlled valve, and the bottom drain servo, which is controlled via a PWM signal. The display and SD card use SPI communication lines, and the keypad uses a set of digital input pins configured with the Arduino Keypad library.

Finally, the pump module holds the three peristaltic pump heads and their tubing. We mounted the pumps in a row inside a 3D-printed enclosure, with their tubing routed through holes to the outside. The pump inlets are connected to the stock solution bottles (water, ethanol, etc.), and the outlets are directed upward into the premix container. We used tubing of different colors (clear vs. blue) to differentiate inlet vs. outlet lines at a glance and marked each line with tape labels indicating its fluid (e.g., "EtOH 100%" or "Water"). This helps ensure correct setup each time, especially if the tubing is removed for cleaning. The pumps were wired to the relay board through long leads that run back to the relay box; we gave the leftmost pump extra lead length so it could reach the far side of the device where the relay box was positioned.

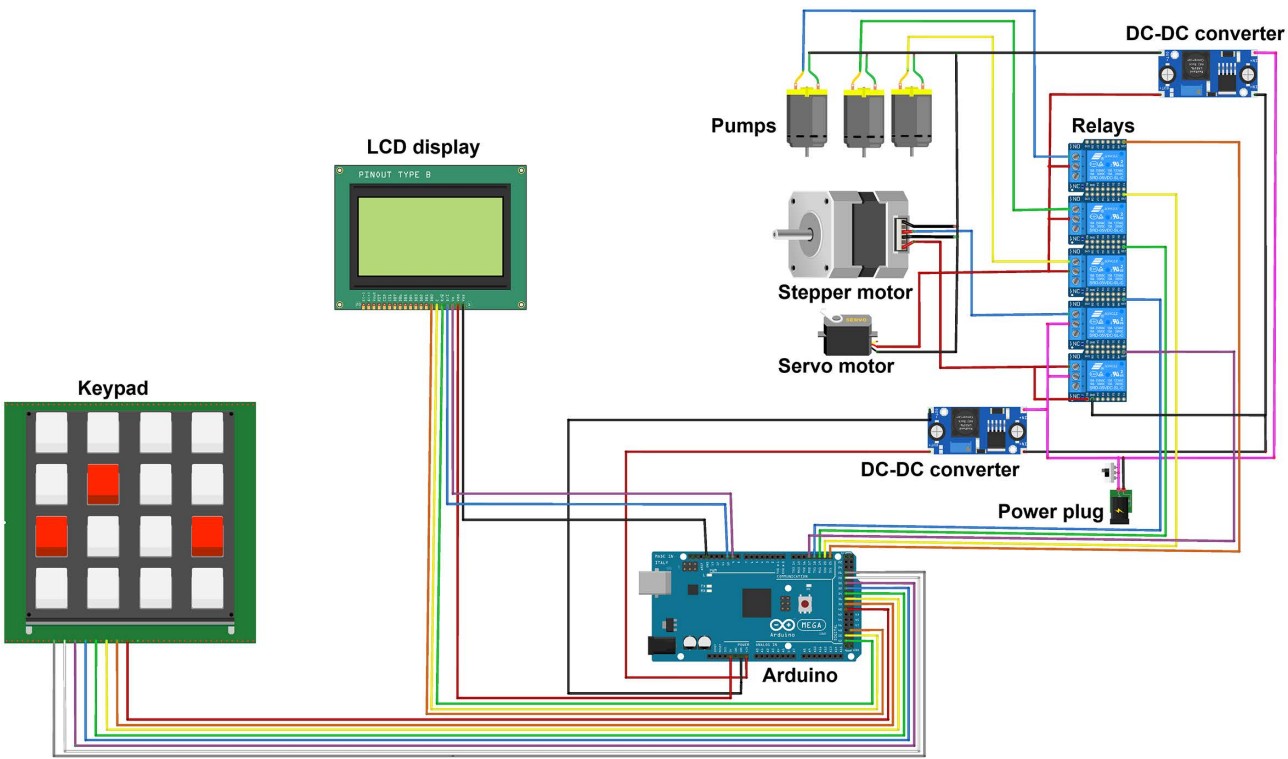

**Fig 3. Schematic diagram of the electrical wiring for the automated tissue processor.** The schematic shows all key components and their interconnections, including the Arduino Mega microcontroller, peristaltic pumps, servo motor, motorized ball valve, relay board, display, keypad, and power supply. Each connection is labeled with corresponding pin numbers as listed in Table 3. The color coding of the wiring is arbitrary and does not indicate functional grouping.

**Table 3. Numbering of pins for electrical wiring.**

| Item | Pin | Item | Pin |
|---|---|---|---|
| Display TFT_CS | 10 | Keypad | [26 –40], even |
| Display TFT_DC | 8 | Servo | 41 |
| Display SD_CS | 11 | DC-DC Converter | 5V & GND |
| Display MOSI | 51 | Relays | 17,18,19,20,21 |
| Display MISO | 50 | Pumps (Water, EtOH96, EtOH100) | 37,35,33 |
| Display SCK | 52 | | |

## Electrical wiring

A schematic of the system's electrical wiring is provided in Fig 3, illustrating the connection layout of all components within the tissue processor. This schematic details how the microcontroller interfaces with actuators, sensors, display units, and user input components. Additionally, Table 3 lists all pin assignments and their respective functions, serving as a reference for wiring verification and troubleshooting.

## Software code

The Arduino code uses several open-source libraries: we leveraged Adafruit's GFX and ST7735 libraries for controlling the TFT display, the Adafruit Keypad library for matrix keypad input, and ArduinoJson for parsing and writing the protocol file on the SD card. These libraries (versions specified in the Supplementary Information) must be installed in the Arduino IDE prior to compiling the code. Aside from these, the code is written in C++ and can be modified by users who wish to add features or change how the menus operate. The code is provided in SI.

## PI Staining protocol

Here, we describe the PI staining procedure applied to all tissue samplesand how it was implemented using the tissue processor. This protocol was adapted from previously published methods [6,8].

Fixation and Initial Storage: We collected fresh plant tissues (approximately 1–2 cm in length) immediately submerged them in FAA fixative (50% ethanol, 5% acetic acid, 3.7% formaldehyde in water). The samples were vacuum-infiltrated in FAA for 1 hour to enhance fixative penetration, then left at 4 °C overnight.

Dehydration/rehydration series. The next day, we transferred the fixed samples to the tissue processor's container, and we ran an ethanol dehydration series. We programmed the processor to follow a sequence of nine steps, each 30 minutes long: 50% ethanol, 70% ethanol, 85% ethanol, 95% ethanol, 100% ethanol, 100% ethanol (repeat), 95% ethanol, 85% ethanol, 70% ethanol, 50% ethanol, 30% ethanol, 15% ethanol, DI water, DI water (repeat). The tissues were left in 70% ethanol overnight as it is a convenient stopping point in the protocol. The two consecutive DI water steps at the end help to wash out any remaining ethanol. By the end of this series, the tissue was fully rehydrated in pure water.

Staining with PI: We then removed the tissue from the tissue container and placed it in a separate container with 2% (w/v) propidium iodide solution (prepared in water). The sample was shaken on an orbital shaker at ~150 rpm for 1 hour in the dark (covered with foil to protect the light-sensitive dye). During this time, PI intercalates into the DNA of cells with permeabilized membranes.

Post-stain Re-dehydration. After staining, we returned the tissue to the tissue container for an ethanol exchange series to remove unbound stain and dehydrate the samples before clearing. We programmed the processor following sequence of ten steps, each 30 minutes long: DI water, DI Water, 15% ethanol, 30% ethanol, 50% ethanol, 70% ethanol, 85% ethanol, 95% ethanol, 100% ethanol, 100% ethanol. By the end of this sequence, the tissue was in absolute ethanol, fully dehydrated and ready for clearing.

Clearing and Mounting: Immediately after the final run, we removed the tissue from the tissue container and placed it in a 1:1 (vol/vol) solution of ethanol: methyl salicylate (a clearing agent with high refractive index). The sample was gently agitated on a shaker for 1 hour in this solution. Following that, we transferred the tissue to 100% methyl salicylate and stored at 4 °C for at least 48 hours. This clearing process renders the tissue more transparent by replacing ethanol in the tissue with methyl salicylate, which reduces light scattering during imaging.

Confocal imaging: We mounted the stained stem segments in 100% methyl salicylate on microscope slides with coverslips. We used confocal laser scanning microscope (Zeiss u880) to image the PI stain, with 514 nm Argon laser line as excitation, and 600–700 nm emission.

## Results

### Design of the automated tissue processor

Fig 1 provides an overview of the processor, showing a labeled schematic diagram (Fig 1a) and a photograph of the assembled system (Fig 1b). The system consists of two main glass containers: a premix container and a tissue container. The tissue sample (e.g., a plant tissue specimen) is placed in the tissue container, where it remains for the duration of processing. The premix container serves as a reservoir where the next solvent solution is prepared at the desired concentration before being transferred into the tissue container. This two-container design allows precise control of concentration changes while keeping the tissue mostly stationary and submerged in a liquid to avoid drying.

We use peristaltic pumps to draw from stock solvent reservoirs—typically distilled water, 70% ethanol, and 100% ethanol—and mix into the premix container to achieve the target concentration. Once the premix container has the correct solution, a motorized ball valve opens to fill the tissue container by gravity-driven flow. After the programmed soaking time, a second valve at the base of the tissue container opens to drain the solution, and the next step begins. This cycle ensures that the tissue remains submerged except during the few seconds required for the drain/refill transition during which it is exposed to air.

We characterized the pump performance by pumping water for fixed time intervals and measuring the volume dispensed using a graduated cylinder. Fig 4 shows experimental measurements of the flow rate of the peristaltic pump, indicating a consistent flow rate of $1.10 \pm 0.02$ mL/s (mean±SD), allowing us to precisely deliver solvent volumes by adjusting pump activation time. For a typical exchange volume of 250 mL, the corresponding delivery time was approximately 4 min, based on the measured pump flow rate, ensuring complete replacement of the tissue container contents well before each a typical 20–30 min incubation step. This approach enabled predictable and repeatable solvent exchange without the need for flow sensors or closed-loop control.

### Software design and operation

The tissue processor is controlled by an Arduino Mega microcontroller running custom software (code provided in the Supplementary Information). To support flexible and user-friendly operation, we developed a simple text-based menu system accessible via keypad and an LCD screen. This interface allows users to configure, modify, and execute solvent exchange protocols directly on the processors. Fig 5 shows a flowchart outlining the structure and available functions within the software interface. Additionally, S1 Video shows step-by-step programming and operation of the tissue processor.

Upon powering the processor, the user is presented with a main menu offering the following functions:

Run Program: Execute a predefined solvent exchange routine. The user selects a *Routine ID* corresponding to a saved protocol, then presses start to run it. During execution, the display shows the current step number, target concentration, elapsed time, and estimated time remaining for the routine. The system automatically steps through all programmed solvent exchanges in sequence, actuating pumps and valves as needed. The user can pause or abort the run via the keypad if required.

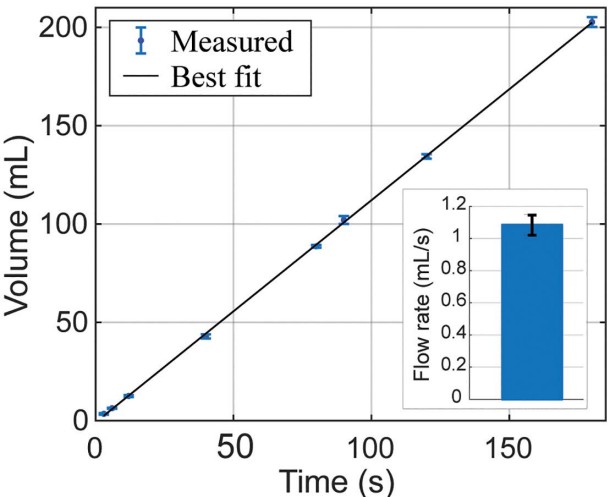

**Fig 4. Experimental measurement of the flow rate of the peristaltic pump.** We operated the pump for fixed time intervals (x-axis) and measured the dispensed liquid volume using a graduated cylinder (y-axis). For each interval, we performed three replicate measurements (n = 3); data points show the mean and error bars indicate the standard deviation (SD). We calculated the flow rate as the ratio of dispensed volume to pump activation time. The inset shows the overall average flow rate calculated from all measurements (n = 24), indicating a flow rate of 1.10 ± 0.02 mL/s (mean ± SD). As summarized in Table 1, the overall system can be assembled for approximately $400, with the main cost contributions coming from the glassware (~$100), peristaltic pumps (~$75), and the microcontroller (~$50).

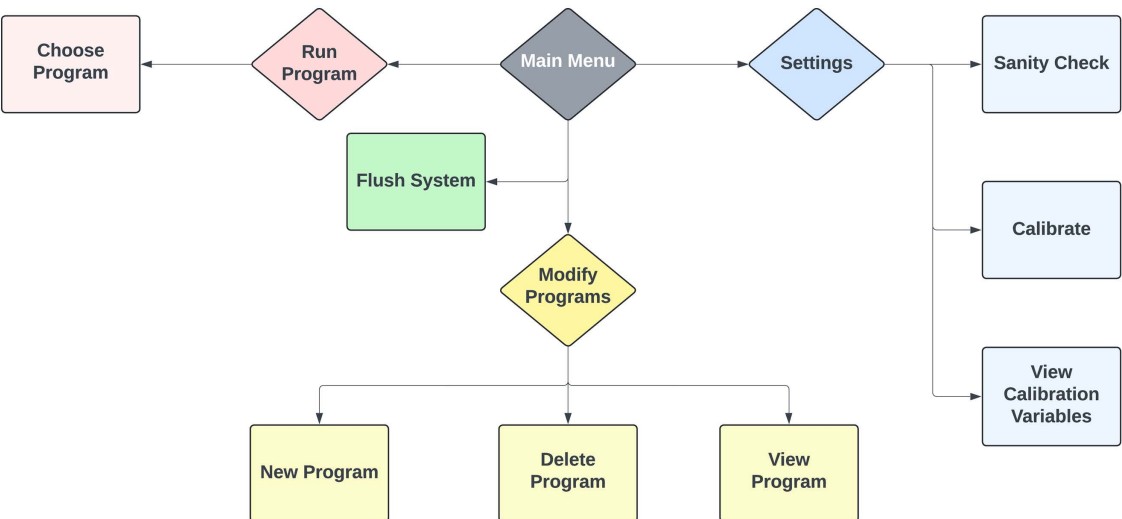

**Fig 5. Flowchart illustrating the software interface and available functions.** The custom Arduino-based program offers a menu-driven user interface for configuring and executing protocols directly on the device. Key functions include running predefined solvent exchange routines, modifying and storing multiple protocols on an SD card, flushing the system for cleaning, and performing diagnostic checks and calibration.

Modify Programs: Create or edit the protocols stored on the SD card. The software organizes protocols in a JSON file on the SD card, which can hold multiple named routines. Through the menu, the user can add a new routine, edit an existing routine's steps, or delete a routine. When editing or creating a protocol, the interface will prompt for each step's duration and solvent concentration. The user navigates through fields using the # key (to accept a value and move forward)

and the * key (to go back), entering numeric values via the keypad. New steps can be added as needed. We found it convenient to program routines directly on the device; however, advanced users may also remove the SD card and edit the JSON file on a PC for quicker text editing of many steps. A View Routine option under this menu allows reviewing the sequence of steps in a selected routine on the screen, so one can verify it before execution.

Flush System: This utility opens all valves and runs all pumps simultaneously to flush the system with clean solvent and empty all lines. It is used to rinse the tubing and containers after a protocol is finished, to ensure no residual chemicals remain before storage or before switching to a different protocol. We typically run the flush procedure at the at the end of a protocol.

Settings | Sanity Check: This diagnostic mode checks that each hardware component is functioning and wired correctly. When selected, the program will sequentially activate each pump, toggle each valve, and read the keypad, providing on-screen feedback (e.g., "Pump 1 OK") for each test. This helps troubleshoot any assembly or connection issues—especially useful the first time the system is built or if any component is replaced.

Settings | Calibrate: This menu guides the user through a calibration of pump flow rates or volumes if needed. In our case, the pumps demonstrated sufficient consistency that a simple time-based dosing approach—calibrated as shown in Fig 4—was adequate for accurate volume delivery. However, the software allows adjusting a scaling factor for delivered volume if, for example, a different pump model is used. There is also a View Calibration option to display the current calibration constants on the screen.

## Performance of tissue processor for staining protocol

To evaluate the processor's performance, we applied it to a standard propidium iodide (PI) staining protocol for tomato (*Solanum lycopersicum*) stem tissues. Identical samples were either processed manually or using the automated system. Fig 6 presents a photograph of the tomato stem immediately after cutting and post-staining, along with confocal microscopy images obtained using manual staining and staining with the tissue processor. Panel (b) shows a fluorescence image of a manually stained sample, while panel (c) displays the intensity profile along the red dashed line indicated in (b). Similarly, panel (d) shows a fluorescence image of a sample processed entirely with the tissue processor, and panel (e) presents the corresponding intensity profile along the red dashed line shown in (d). Panel (f) summarizes the signal-to-noise ratios calculated from these intensity profiles for multiple independently processed samples (*n*=3), showing consistently high values (~80) for both manual and automated staining. This demonstrates that the automated processor reliably reproduces manual results while significantly reducing hands-on time.

For the PI staining workflow presented here, the processor executed more than 20 sequential solvent exchange steps, each with a programmed duration of 30 min, resulting in a total automated processing time of approximately 10 hours. User hands-on time was limited to protocol programming and solvent loading, requiring less than ~10 minutes per run, compared to several hours of intermittent manual intervention for hand-processed samples.

To demonstrate the versatility of the staining process using the tissue processor, we applied it to PI-stain various plant species. Fig 7 shows PI staining of a fig leaf (*Ficus carica*) (a), a tomato leaf (*Solanum lycopersicum*) (b), an *Euphorbia peplus* leaf (c), and a cross-section of an *Arabidopsis thaliana* flower (d). All tissues were subsequently cleared with methyl salicylate and imaged on a Zeiss LSM 880 confocal microscope (514 nm excitation, 600–700 nm emission). Throughout the PI staining protocol, the processor executed more than 20 solvent exchange steps—including dehydration, rehydration, and post-stain washes—ensuring consistent solvent exposure and precise timing. By keeping tissues submerged and automating each step, the device minimized drying artifacts and user-induced variability. These results demonstrate that the processor provides a reliable, reproducible solution for multistep staining workflows, significantly reducing hands-on labor while maintaining experimental quality.

## Discussion

### Design consideration

We designed the processor to balance cost, ease of fabrication, and reliable operation. We selected peristaltic pumps because they are self-priming and prevent cross-contamination. Moreover, they are reliable for repetitive liquid handling, with routine maintenance limited to periodic tubing replacement.

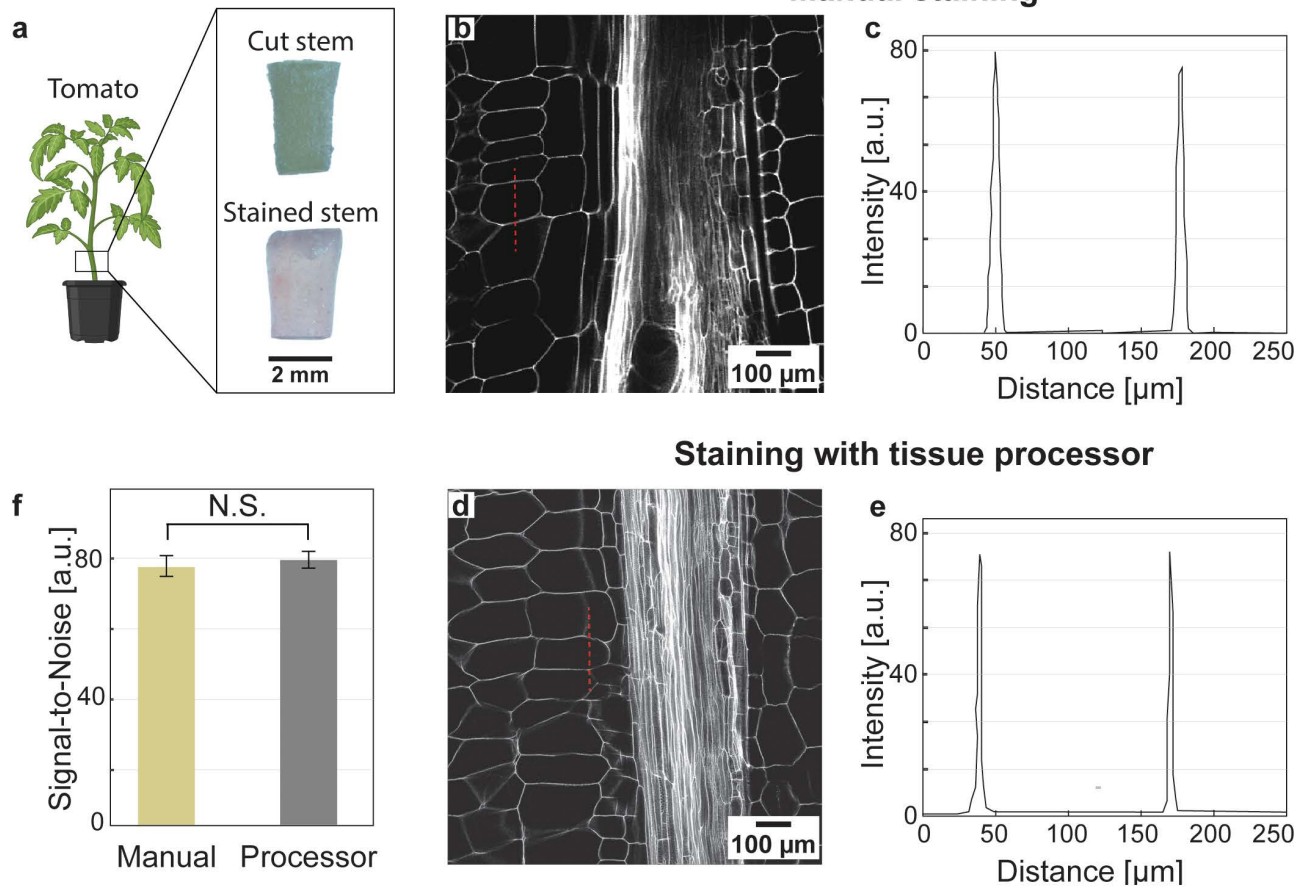

**Fig 6. Tomato (*Solanum lycopersicum*) stem tissues stained with propidium iodide (PI). (a)** Photographs of tomato stems excised from 4-week-old plants, shown immediately after cutting and following staining with PI. **(b)** Fluorescence image of a manually stained sample. **(c)** Intensity profile along the red dashed line shown in **(b)**. **(d)** Fluorescence image of a sample processed entirely using the automated tissue processor. **(e)** Intensity profile along the red dashed line shown in **(d)**. **(f)** Comparison of signal-to-noise ratios (SNR), defined as the mean intensity of the stained cell wall divided by background intensity, calculated from the profiles in (c) and **(e)**. Both manual and staining with the tissue processor yield similarly high SNR values (~80), with no significant difference between methods (t-test, $P > 0.05$; $n = 3$ tissue samples processed in three independent runs), indicating comparable staining quality. The processor performed over 20 solvent exchanges—including dehydration, rehydration, and post-stain washes—with minimal manual handling. All samples were cleared with methyl salicylate and imaged on a Zeiss 880 confocal microscope (514 nm excitation, 600–700 nm emission).

To transfer solvent from the premix to the tissue container, we used gravity-driven flow controlled by motorized valves instead of additional pumps, as this step does not require precise volume delivery.

Solvents primarily contact the glass containers and the tubing that carries solutions from stock bottles to the premix container. There is no direct solvent exposure during operation; however, appropriate care should be taken when filling solvent containers and emptying the waste bottle. If toxic solvents are used, the system can be operated inside a chemical fume hood due to its compact size. We used silicone tubing, which is compatible with ethanol and a wide range of polar solvents. However, users should avoid non-polar solvents like toluene or xylene, as silicone degrades under prolonged exposure to such liquids; in such cases, chemically resistant alternatives such as PTFE tubing should be used.

We opted for an Arduino-based control system paired with a keypad and display, which enables standalone programming of custom protocols and is widely supported within the maker and research communities. In its current

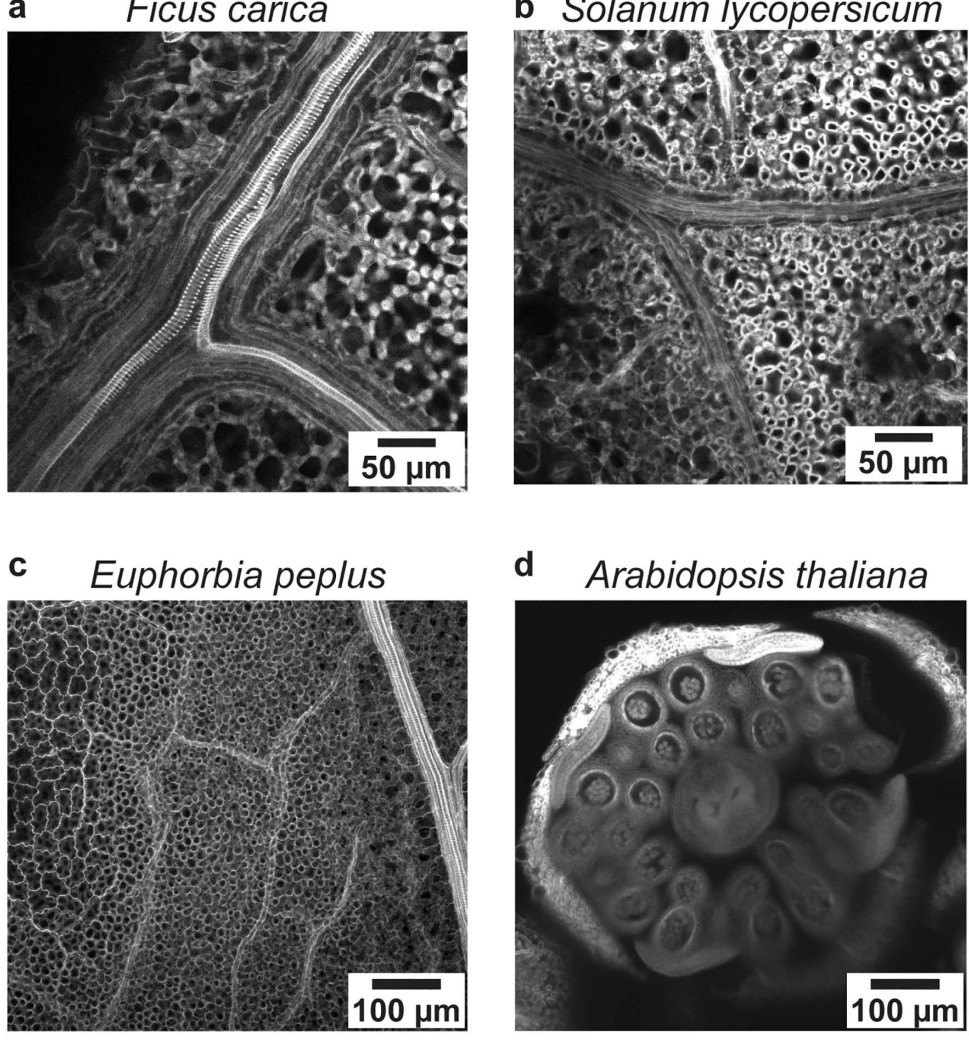

**Fig 7. Confocal images of plant tissues stained using the automated tissue processor.** Representative samples include: **(a)** *Ficus carica* leaf, **(b)** *Solanum lycopersicum* leaf, **(c)** *Euphorbia peplus* leaf, and **(d)** cross-section of a flower from *Arabidopsis thaliana*. Tissues were stained with propidium iodide (PI), cleared with methyl salicylate, and imaged using a Zeiss LSM 880 confocal microscope (514 nm excitation, 600–700 nm emission).

configuration, the processor accommodates a 300 mL tissue container, allowing multiple tissue samples to be processed simultaneously within the container. To increase sample number or size, a larger tissue container can be used.

## Limitation of the tissue processor

While the tissue processor reliably automates multistep solvent exchange protocols and reproduces manual staining quality, several limitations should be acknowledged. First, we did not perform a direct comparison with commercial tissue processors due to their high cost and limited accessibility in many laboratories, including ours. Manual staining remains the most used approach; therefore, comparison to manual staining provides a relevant and practical benchmark for the intended user base of the tissue processor presented here. Future work could extend this comparison to commercial tissue processors where access is available.

Second, the current implementation of the tissue processor does not include closed-loop sensing (for example, flow or liquid-level sensors) to autonomously detect errors such as pump failure or valve blockage. However, the tissue processor described here has been used routinely in our laboratory for over one year without observed hardware or software failures. Incorporating feedback mechanisms for error handling would further improve robustness but would also increase system complexity and cost. We therefore prioritized low cost and ease of replication in the present design, and note that the integration of closed-loop sensing is beyond the scope of the current work.

## Conclusion

We developed a low-cost, open-source tissue processor for automated solvent exchange in biological sample preparation. Built from accessible components and equipped with programmable software, the processor enables solvent exchange with minimal user intervention. As a demonstration, we used the processor to perform a propidium iodide staining protocol, automating over 20 sequential dehydration and rehydration steps. We tested the protocol on multiple plant species and tissue morphologies, including leaves, stems, and flowers. These data were collected across different users and independent processing batches, highlighting the processor's versatility and reproducibility. Throughout the protocol, the processor maintained consistent timing, which reduced user workload from hours to just a few minutes, including programming the tissue processor and filling the solvent bottles. Unlike commercial tissue processors—often expensive and overengineered for standard lab needs—our design is affordable (~\$400), compact, and easy to replicate using parts commonly found in makerspaces. The processor is not limited to hydration or dehydration steps; it can be readily adapted for protocols requiring stepwise solution changes, including staining, clearing, fixation, or washing. We note that we used ethanol as the solvent in this work; solvent compatibility depends on the tubing material and should be considered for other protocols (see Design Considerations). We envision that this processor could be readily used for animal tissues and could be further improved by incorporating closed-loop sensing and feedback for better error handling (see Limitations of the tissue processor).

## Supporting information

**S1 Data. This file contains the software code used to operate the tissue processor, the numerical data used to generate the plot in Fig 4, the values underlying the mean and standard deviation reported in Fig 6f, and the data points extracted from image intensity profiles used for the analyses shown in Fig 6c and Fig 6e.** The CAD files, provided in both STEP and SLDPRT formats, are available in the GitHub repository: https://doi.org/10.5281/zenodo.17050743.
(DOCX)

**S1 Video. Operation of tissue processor.** This video shows step-by-step programming and operation of the tissue processor.
(MP4)

## Acknowledgments

We thank A. Johnson, A. Bao, M. Heeney, A. Jung, A. Roeder, S. Yanders and E. Boisvert for providing plant samples, and images of plant samples processed with the tissue processor. The PI staining protocol was modified from a protocol developed by Sam Leiboff.

## Author contributions

**Conceptualization:** May Wang, Samantha Pelletier, Anand Kumar Mishra, Abraham D. Stroock, Vesna Bacheva.
**Funding acquisition:** Robert F. Shepherd, Margaret H. Frank, Abraham D. Stroock, Vesna Bacheva.

**Investigation:** Anand Kumar Mishra, Vesna Bacheva.

**Methodology:** May Wang, Samantha Pelletier, Alexis Ellis, Vesna Bacheva.

**Software:** May Wang, Vesna Bacheva.

**Supervision:** Robert F. Shepherd, Margaret H. Frank, Abraham D. Stroock, Anand Kumar Mishra, Vesna Bacheva.

**Validation:** Samantha Pelletier, Vesna Bacheva.

**Writing – original draft:** Vesna Bacheva.

**Writing – review & editing:** Vesna Bacheva with input from all co-authors.

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
