## [Decision Letter · Decision Letter 0]

3 Nov 2025

Dear Dr. Vesna Bacheva,

Thank you for submitting your manuscript to PLOS ONE. After careful consideration, we feel that it has merit but does not fully meet PLOS ONE’s publication criteria as it currently stands. Therefore, we invite you to submit a revised version of the manuscript that addresses the points raised during the review process.

We look forward to receiving your revised manuscript.

Kind regards,

Fentahun Adane Nigat, MSc., PhD

Academic Editor

PLOS ONE

Journal Requirements:

“National Science Foundation STC Center for Research on Programmable Plant System under grant number DBI-2019674.

V. Bacheva was supported by Schmidt Science Fellows, SNF Postdoc.Mobility (grant number 214477), and KIC Postdoctoral Fellowship”

“National Science Foundation STC Center for Research on Programmable Plant System under grant number DBI-2019674.

V. Bacheva was supported by Schmidt Science Fellows, SNF Postdoc.Mobility (grant number 214477), and KIC Postdoctoral Fellowship”

“This project was supported by the National Science Foundation STC Center for Research on Programmable Plant System under grant number DBI-2019674. V. Bacheva was supported by Schmidt Science Fellows, SNF Postdoc.Mobility (grant number 214477), and KIC Postdoctoral Fellowship”

“National Science Foundation STC Center for Research on Programmable Plant System under grant number DBI-2019674.

V. Bacheva was supported by Schmidt Science Fellows, SNF Postdoc.Mobility (grant number 214477), and KIC Postdoctoral Fellowship”

Additional Editor Comments (if provided):

Thank you for submitting your manuscript entitled “Low-cost, DIY programmable tissue processor for solvent exchange in biological sample preparation” to PLOS ONE. The reviewers and I recognize the novelty and practical relevance of your work, which presents an innovative and potentially impactful approach to affordable, open-source laboratory automation. However, several critical issues must be addressed before the manuscript can be reconsidered for publication.

Manuscript structure and organization

The current version does not fully adhere to the journal’s required structure. Please revise the manuscript to include clearly defined sections—Introduction, Materials and Methods, Results, Discussion, and Conclusion—with logical flow and sufficient methodological detail. Ensure that limitations are clearly discussed.

Title and scope

Revise the title to better reflect the evaluative and experimental nature of the study. For example, a title such as “Evaluation of a Low-Cost, DIY Programmable Tissue Processor for Solvent Exchange in Biological Sample Preparation” would more accurately convey the focus on device performance.

Validation of the software and hardware

Provide experimental validation data for both the control software and hardware system. This should include:

Software reliability: accuracy of timing and sequencing, error-handling capacity, and reproducibility.

Hardware performance: comparison with a commercial or standard tissue processor in terms of tissue quality, staining uniformity, and solvent exchange efficiency.

Quantitative results and appropriate statistical analyses should be presented to support the device’s claimed performance.

Quantitative performance assessment

Include clear performance metrics such as processing time, solvent exchange rate, and reproducibility across multiple runs. Comparative data will strengthen the technical credibility of the device.

Statistical and experimental details

Please specify the number of replicates (n), the statistical tests used, and any measures of variability (e.g., mean ± SD). These additions are necessary to validate reproducibility and reliability.

Safety, scalability, and reproducibility

Add a section discussing safety considerations (e.g., solvent vapor management) and the device’s scalability, including its ability to process multiple samples simultaneously. Mention expected lifespan and maintenance requirements for key components.

Context and literature

Expand the Introduction to include comparative context with other open-source automation tools (e.g., OpenTrons, microfluidic systems). Clarify the unique innovation of your design (e.g., whether it automates both dehydration and rehydration steps).

Figures, tables, and data presentation

Ensure all figures have complete legends, proper scale bars, and consistent numbering. Tables should include total cost estimates and availability of components. Verify that all figures are cited sequentially in the text.

Language, formatting, and references

Conduct a thorough grammatical and editorial revision to improve clarity and professional tone. Ensure consistent formatting of units, symbols, and reference style, and include DOIs for all references where possible.

Additional scientific context

As noted by one reviewer, please mention in the Introduction that imaging techniques are used not only for examining cell growth and responses but also for assessing cell death, to provide a more comprehensive context for the use of tissue imaging.

Once these revisions are completed, please provide:

A revised manuscript with all corrections clearly highlighted or tracked, and

A detailed point-by-point response addressing each reviewer comment and editorial note.

We believe that, with substantial revision and additional validation, this study could make a meaningful contribution to the field of open-source biomedical instrumentation.

Reviewers' comments:

Reviewer's Responses to Questions

**Comments to the Author**

1. Is the manuscript technically sound, and do the data support the conclusions?

Reviewer #1: Yes

Reviewer #2: Yes

Reviewer #3: Yes

2. Has the statistical analysis been performed appropriately and rigorously?

Reviewer #1: No

Reviewer #2: No

Reviewer #3: N/A

3. Have the authors made all data underlying the findings in their manuscript fully available?

Reviewer #1: Yes

Reviewer #2: Yes

Reviewer #3: Yes

4. Is the manuscript presented in an intelligible fashion and written in standard English?

Reviewer #1: No

Reviewer #2: Yes

Reviewer #3: Yes

Reviewer #1: Manuscript title: Low-cost, DIY programmable tissue processor for solvent exchange in biological sample preparation

General comments: Thank you for the opportunity to review this manuscript, which presents a novel and potentially valuable low-cost, DIY programmable tissue processor. The concept is highly relevant, as accessible and affordable instrumentation can significantly advance biological and medical researches. However, in its current form, the manuscript requires major revision before it can be considered for publication. The main drawbacks are lack of structural rigor and insufficient validation data. The following major flaws must be addressed to meet the scientific standards expected by this journal.

1. The current title is descriptive but does not accurately reflect the evaluative nature of the work that is required for a scientific publication. It should be revised to emphasize the performance assessment of the device. Consider a title such as ‘Evaluation of a Low-Cost, DIY Programmable Tissue Processor for Solvent Exchange in Biological Sample Preparation’.

2. The manuscript does not adhere to the standard structure (e.g., Introduction, Methods, Results, Discussion, Conclusion) required by the journal. The absence of clearly defined sections for Methods, Results, and a Discussion that without the study limitations makes it difficult to assess the validity and scope of the work. The manuscript must be thoroughly reorganized to meet these fundamental publication standards.

3. A critical omission is the validation of the custom-developed software. The manuscript should include data demonstrating the software's reliability, such as Accuracy- Does the software execute the programmed timing and sequence steps correctly and consistently? Error Handling - How does the software manage user input errors or hardware communication failures? Reproducibility- Are the programmed protocols reproducible across multiple runs and by different users?

4. The study lacks a quantitative performance evaluation of the hardware itself. To establish the device's utility, it is essential to compare its performance against a commercial or standard benchmark. This analysis should include: - Sensitivity- can the device consistently and effectively process a range of tissue types and sizes to a high-quality standard? Specificity/Robustness- are the results free from artifacts introduced by the device? Data comparing tissue morphology, staining quality, and processing consistency (e.g., against a commercial processor) is necessary. Performance Metrics- include data on processing time, solvent exchange efficiency, and consistency of results across multiple batches.

5. The authors are strongly encouraged to perform a comprehensive revision of the manuscript, ensuring it fully aligns with all specific formatting and content guidelines for the journal (PLOS ONE). This includes the incorporation of all standard scientific manuscript sections, a clear description of limitations, and the critical validation data outlined above.

Reviewer #2: Comments to the authors:

Title: “Low-cost, DIY programmable tissue processor for solvent exchange in biological sample preparation”

Manuscript Number: (PONE-D-25-48079)

Dear the editor of PLOS ONE Journal

I would like to express my sincere gratitude for the opportunity to review this manuscript and contribute my expertise to the advancement of scientific knowledge. I also wish to extend my appreciation to the authors for their valuable contribution in exploring an innovative and time-efficient technology for tissue processing. Their work addresses a significant need in histological research and diagnostics by proposing a method that has the potential to reduce the time, labor, and resource demands traditionally associated with conventional tissue processing procedures.

1. Title: “Low-cost, DIY programmable tissue processor for solvent exchange in biological sample preparation” (PONE-D-25-48079):

• Comment: The title is concise, descriptive, and captures the core innovation of the work “low-cost,” “DIY,” and “programmable” are strong keywords that attract interest in open-source hardware and biological sample preparation.

• Suggestions:

o Consider specifying the type of tissues or applications (e.g., “for histological and fluorescence microscopy sample preparation”) to make the scope clearer.

o Describe the “DIY” first and make formal contexts since journals like PLOS ONE prefer academic tone.

2. Abstract

• Comment: The abstract clearly summarizes the motivation, approach, design, and findings. It effectively emphasizes accessibility and cost reduction.

• Suggestions:

o The problem statement could briefly include quantitative efficiency (e.g., percentage reduction in manual time).

o Add a brief mention of validation results (e.g., comparable fluorescence intensity to manual staining).

3. Introduction

• Comment: The introduction is well structured, providing a logical flow from biological imaging needs to limitations of existing systems and the motivation for a low-cost alternative. References (1–16) are appropriate and current.

• Suggestions:

o Include a comparative statement with similar open-source automation tools (e.g., OpenTrons or other microfluidic DIY devices).

o Add a clear hypothesis or research question at the end of the section (e.g., “We hypothesize that a low-cost programmable tissue processor can achieve staining results equivalent to commercial systems”).

o Clarify the novelty, does the processor uniquely automate both dehydration and rehydration steps or simply adapt existing robotics principles?

o Check citation consistency (e.g., space before parentheses: “subsequent tissue clearing(6)” → “subsequent tissue clearing (6)”).

4. Design of the Automated Tissue Processor

• Comment: This section provides excellent technical clarity. Figures 1–2 support understanding.

• Suggestions:

o Include a brief cost breakdown within this section, not only in Table 1, to strengthen the claim of affordability.

o A risk assessment or safety note (e.g., ethanol vapor management) could improve completeness.

o Clarify scalability, can this handle multiple samples simultaneously?

o Add comparison with existing low-cost automation platforms for context (e.g., Arduino-controlled microfluidic systems).

5. Software Design and Operation

• Comment: Comprehensive description of the user interface and operational logic. The inclusion of a flowchart (Fig. 3) is excellent.

• Suggestions:

• Indicate whether the software includes error handling or alerts (e.g., empty solvent reservoirs).

• Consider including code versioning or repository DOI earlier for transparency.

• A short subsection on usability testing (who tested it, user feedback) could strengthen reproducibility claims.

6. Performance of the Tissue Processor

• Comment: Validation using propidium iodide (PI) staining on multiple plant species is appropriate and convincing. Quantitative fluorescence comparison (signal-to-noise ratio ~80) supports equivalency.

• Suggestions:

o Include statistical analysis (e.g., t-test comparing mean intensity between manual and automated samples).

o Provide replicate numbers (n) for reproducibility.

o Add discussion on long-term performance or maintenance stability (e.g., after repeated use).

o Minor: specify microscope model consistently (e.g., “Zeiss LSM 880 confocal microscope” should appear consistently, not “Zeiss u880”).

7. Design Considerations

• Comment: Good rationale for material and component selection.

• Suggestions:

o Add a table of solvent compatibility for quick reference.

o Include expected lifespan or maintenance recommendations for pumps and valves.

o Clarify whether temperature or humidity control is needed for reliable operation.

o Mention possible future modifications (e.g., incorporation of sensors or remote data logging).

8. Conclusion

• Comment: The conclusion effectively summarizes the study’s achievements and potential applications.

• Suggestions:

o Add a quantitative summary (e.g., cost reduction, processing time, and staining reproducibility).

o End with a forward-looking statement about potential expansion to animal tissues or integration with automated imaging systems. Example: “Future work could extend this platform to integrate real-time imaging or temperature-controlled sample handling.”

9. Materials and Methods

• Comment: Extremely detailed and well written, especially for reproducibility. The use of open-source files and a GitHub repository aligns with PLOS ONE’s data policy.

• Suggestions:

o For clarity, include subheadings under “Assembly Procedure” (e.g., “Control Box,” “Pump Module”).

o Add a short note on ethical or biosafety considerations if using plant materials.

o Ensure units and symbols follow consistent formatting (e.g., “°C,” “μm”).

o In “Electrical wiring,” specify any safety isolation between power and control circuits.

o Table 1 could include total estimated cost and country availability of parts.

10. Figures and Tables

• Comment: Figures are clear and informative. Table 1–3 provide high transparency.

• Suggestions:

o Ensure all figures include scale bars for microscopy images.

o Figures 1 and 7: add legends with abbreviations for clarity.

o Combine Fig. 2 (pump calibration) and a small chart comparing manual vs. automated results for visual impact.

o Verify all figure numbers are cited in the text sequentially.

11. References

• Comment: The reference list is recent, relevant, and includes both foundational and contemporary studies (e.g., PLOS ONE 2024 papers).

• Suggestions:

o Ensure reference consistency (some journal names italicized, others not).

o Add DOIs for all references where possible.

o A few references (e.g., “Tissue Processing Systems Market...”) are non-peer-reviewed and should be supported by primary sources if available. The status of the paper is ongoing.

o Include one or two recent citations (2023–2025) on open-source lab automation to contextualize innovation.

Overall Assessment

• Strengths:

o High reproducibility and transparency (open-source design, detailed methods).

o Excellent integration of engineering and biological applications.

o Figures and methods are well structured and clear.

• Weaknesses & Improvements:

o Limited statistical validation and quantitative comparison.

o Some redundancy across sections (especially Abstract vs. Conclusion).

o “DIY” phrasing may reduce perceived technical rigor.

o Lack of discussion on safety, error handling, and scalability.

Reviewer #3: Imaging techniques are not only used to examine the cell growth and responses but also to examine cell death. Therefore, please add this information under the introduction part. Also work on the grammatical and editorial corrections.

**Do you want your identity to be public for this peer review?** For information about this choice, including consent withdrawal, please see our Privacy Policy

Reviewer #1: No

Reviewer #2: **Yes:** Dr. Hussen Abdu Muhidin

Reviewer #3: No

---

## [Author Response · Author response to Decision Letter 1]

21 Dec 2025

Editor

The editor notes that: “Thank you for submitting your manuscript entitled “Low-cost, DIY programmable tissue processor for solvent exchange in biological sample preparation” to PLOS ONE. The reviewers and I recognize the novelty and practical relevance of your work, which presents an innovative and potentially impactful approach to affordable, open-source laboratory automation. However, several critical issues must be addressed before the manuscript can be reconsidered for publication.”

We thank the editor for their specific comments and corrections (bold font). Here we provide a point-by-point response (regular font) on our modifications and corrections. Revised sections in the manuscript and supplementary information file are marked in blue font.

1. Manuscript structure and organization

The current version does not fully adhere to the journal’s required structure. Please revise the manuscript to include clearly defined sections—Introduction, Materials and Methods, Results, Discussion, and Conclusion—with logical flow and sufficient methodological detail. Ensure that limitations are clearly discussed.

We have revised the manuscript to include all required sections: Introduction, Materials and Methods, Results, Discussion, and Conclusion. The Materials and Methods section is organized into subsections describing each method and material used. The Results section is structured into subsections corresponding to the different experimental results, including the design and evaluation of the tissue processor. The Discussion includes subsections on design considerations, as well as a newly added section addressing the limitations of the system.

2. Title and scope

Revise the title to better reflect the evaluative and experimental nature of the study. For example, a title such as “Evaluation of a Low-Cost, DIY Programmable Tissue Processor for Solvent Exchange in Biological Sample Preparation” would more accurately convey the focus on device performance.

We revised the title to “Design and evaluation of a low-cost, DIY programmable tissue processor for solvent exchange in biological sample preparation”, reflecting that the work both presents the design of the processor and evaluates its performance.

3. Validation of the software and hardware

Provide experimental validation data for both the control software and hardware system. This should include:

-Software reliability: accuracy of timing and sequencing, error-handling capacity, and reproducibility.

-Hardware performance: comparison with a commercial or standard tissue processor in terms of tissue quality, staining uniformity, and solvent exchange efficiency.

-Quantitative results and appropriate statistical analyses should be presented to support the device’s claimed performance.

We now include statistical analyses of tissue staining quality, based on multiple biological replicates. As a quantitative metric, we report the signal-to-noise ratio of stained tissue obtained from manual staining versus staining performed with the tissue processor. This data is presented in the updated Fig. 4.

Our primary motivation for developing this tissue processor is the high cost and limited accessibility of commercial tissue processors, which are not available in many laboratories, including ours. We note that manual staining remains the most commonly used approach; therefore, comparison with manual staining is both relevant and appropriate for the intended user base of this device. We have clarified this rationale in the manuscript and note that future work could extend this comparison to commercial tissue processors where access is available. In addition, we have added a new subsection addressing the limitations of error handling in the present implementation. These limitations are now discussed in the newly added “Limitations of the tissue processor” section of the Discussion.

5. Quantitative performance assessment

Include clear performance metrics such as processing time, solvent exchange rate, and reproducibility across multiple runs. Comparative data will strengthen the technical credibility of the device.

As noted in our response to Comment 3, we have now added quantitative statistical analysis of the signal-to-noise ratio (SNR) in Fig. 4, comparing manual staining with automated tissue processing across multiple biological replicates. These data show no significant difference between the two methods, demonstrating reproducible performance across independent runs.

In addition, we have included a new paragraph summarizing the total processing time of the staining workflow, as well as a discussion on the solvent exchange rate. These additions appear in the sections Design of the automated tissue processor and Performance of tissue processor for staining protocol.

6. Statistical and experimental details

Please specify the number of replicates (n), the statistical tests used, and any measures of variability (e.g., mean ± SD). These additions are necessary to validate reproducibility and reliability.

We have added the missing statistical details throughout the manuscript. Specifically, in Fig.2, we specify the number of replicates used to characterize the pump flow rate. In Fig. 4, we report the number of biological replicates used for the signal-to-noise ratio comparison and specify the statistical test applied.

7. Safety, scalability, and reproducibility

Add a section discussing safety considerations (e.g., solvent vapor management) and the device’s scalability, including its ability to process multiple samples simultaneously. Mention expected lifespan and maintenance requirements for key components.

We have expanded the Design Considerations section to address solvent safety, the expected lifespan and maintenance of key components, as well as volume of tissue samples that can be simultaneously processed and how the system can be scaled.

8. Context and literature

Expand the Introduction to include comparative context with other open-source automation tools (e.g., OpenTrons, microfluidic systems). Clarify the unique innovation of your design (e.g., whether it automates both dehydration and rehydration steps).

We expanded the Introduction to place our work in the context of existing open-source automation tools and clarified that existing liquid-handling platforms and microfluidic systems are not designed for processing centimeter-scale tissues, and highlighted that our design is a low-cost, open-source solution for automated solvent exchange at this scale.

9. Figures, tables, and data presentation

Ensure all figures have complete legends, proper scale bars, and consistent numbering. Tables should include total cost estimates and availability of components. Verify that all figures are cited sequentially in the text.

We have verified that all figures and tables are properly cited and presented.

10. Language, formatting, and references

Conduct a thorough grammatical and editorial revision to improve clarity and professional tone. Ensure consistent formatting of units, symbols, and reference style, and include DOIs for all references where possible.

We revised the manuscript for clarity and consistency. These edits are highlighted in blue font in the revised submission.

11. Additional scientific context

As noted by one reviewer, please mention in the Introduction that imaging techniques are used not only for examining cell growth and responses but also for assessing cell death, to provide a more comprehensive context for the use of tissue imaging.

We have revised the Introduction to explicitly note that imaging techniques also used to study cell death.

Reviewer 1

The reviewer notes that: “Thank you for the opportunity to review this manuscript, which presents a novel and potentially valuable low-cost, DIY programmable tissue processor. The concept is highly relevant, as accessible and affordable instrumentation can significantly advance biological and medical researches. However, in its current form, the manuscript requires major revision before it can be considered for publication. The main drawbacks are lack of structural rigor and insufficient validation data. The following major flaws must be addressed to meet the scientific standards expected by this journal.”

We thank the reviewer for their specific comments and corrections (bold font). Here we provide a point-by-point response (regular font) on our modifications and corrections. Revised sections in the manuscript and supplementary information file are marked in blue font.

1. The current title is descriptive but does not accurately reflect the evaluative nature of the work that is required for a scientific publication. It should be revised to emphasize the performance assessment of the device. Consider a title such as ‘Evaluation of a Low-Cost, DIY Programmable Tissue Processor for Solvent Exchange in Biological Sample Preparation’.

Following the reviewer’s guidance, we revised the title to “Design and evaluation of a low-cost, DIY programmable tissue processor for solvent exchange in biological sample preparation”, reflecting that the work both presents the design of the processor and evaluates its performance.

2. The manuscript does not adhere to the standard structure (e.g., Introduction, Methods, Results, Discussion, Conclusion) required by the journal. The absence of clearly defined sections for Methods, Results, and a Discussion that without the study limitations makes it difficult to assess the validity and scope of the work. The manuscript must be thoroughly reorganized to meet these fundamental publication standards.

We have revised the manuscript to include all required sections: Introduction, Materials and Methods, Results, Discussion, and Conclusion. The Materials and Methods section is organized into subsections describing each method and material used. The Results section is structured into subsections corresponding to the different experimental results, including the design and evaluation of the tissue processor. The Discussion includes subsections on design considerations, as well as a newly added section addressing the limitations of the system.

3. A critical omission is the validation of the custom-developed software. The manuscript should include data demonstrating the software's reliability, such as Accuracy- Does the software execute the programmed timing and sequence steps correctly and consistently? Error Handling - How does the software manage user input errors or hardware communication failures? Reproducibility- Are the programmed protocols reproducible across multiple runs and by different users?

We have now added statistical analyses of tissue staining quality based on multiple biological replicates. As a quantitative metric, we report the signal-to-noise ratio (SNR) of stained tissue obtained using manual staining versus staining performed with the tissue processor; these data are presented in the updated Fig. 4. These results suggest that the tissue processor operates reproducibly across multiple runs and different users, and that the software executes the programmed protocols accurately, as no device-induced artifacts were observed. Moreover, characterization of the pump flow rate (Fig. 2) further demonstrates the robustness of the pump system, which is critical for achieving the correct solvent concentrations.

Regarding error handling, the software includes a built-in diagnostic program (Sanity Check) that verifies whether each hardware component is functioning and wired correctly prior to operation. However, the current implementation of the tissue processor does not include closed-loop sensing (for example, flow or liquid-level sensors) to autonomously detect errors such as pump failure or valve blockage. To address this point, we added a new subsection titled “Limitations of the tissue processor” in the Discussion, where the limitations of error handling in the present implementation are explicitly discussed.

4. The study lacks a quantitative performance evaluation of the hardware itself. To establish the device's utility, it is essential to compare its performance against a commercial or standard benchmark. This analysis should include: - Sensitivity- can the device consistently and effectively process a range of tissue types and sizes to a high-quality standard? Specificity/Robustness- are the results free from artifacts introduced by the device? Data comparing tissue morphology, staining quality, and processing consistency (e.g., against a commercial processor) is necessary. Performance Metrics- include data on processing time, solvent exchange efficiency, and consistency of results across multiple batches.

As noted in the response to Comment 3, we have now added quantitative statistical analyses of the signal-to-noise ratio across multiple biological replicates, and between manual staining with automated tissue processing, shown in Fig. 4. These data show no significant difference between the two methods, demonstrating reproducible performance across independent runs. In addition, Fig. 5 demonstrates the processor’s ability to handle a range of tissue morphologies, sizes, and species, including leaves, stems, and floral tissues from multiple plant species. These data were collected across different users and independent processing batches, indicating robust and reproducible performance. We now explicitly discuss this robustness in the Conclusion section.

In addition, we have included a new paragraph summarizing the total processing time of the staining workflow, as well as a discussion on the solvent exchange rate. These additions appear in the sections Design of the automated tissue processor and Performance of tissue processor for staining protocol.

Our primary motivation for developing this tissue processor is the high cost and limited accessibility of commercial tissue processors, which are not available in many laboratories, including ours. Manual staining remains the most commonly used approach; therefore, comparison with manual staining provides a relevant and appropriate benchmark for the intended user base of this device. Importantly, we observe no staining artifacts introduced by the tissue processor relative to manual processing. We clarify this rationale in the manuscript and note that future work could extend these comparisons to commercial tissue processors where access is available.

5. The authors are strongly encouraged to perform a comprehensive revision of the manuscript, ensuring it fully aligns with all specific formatting and content guidelines for the journal (PLOS ONE). This includes the incorporation of all standard scientific manuscript sections, a clear description of limitations, and the critical validation data outlined above.

We revised the manuscript to align with PLOS ONE guidelines and incorporated the requested clarifications, limitations, and validation.

Reviewer 2

The reviewer notes that: “I would like to express my sincere gratitude for the opportunity to review this manuscript and contribute my expertise to the advancement of scientific knowledge. I also wish to extend my appreciation to the authors for their valuable contribution in exploring an innovative and time-efficient technology for tissue processing. Their work addresses a significant need in histological research and diagnostics by proposing a method that has the potential to reduce the time, labor, and resource demands traditionally associated with conventional tissue processing procedures.”

We thank the reviewer for their specific comments and corrections (bold font). Here we provide a point-by-point response (regular font) on our modifications and corrections. Revised sections in the manuscript and supplementary information file are marked in blue font.

1. Title: “Low-cost, DIY programmable tissue processor for solvent exchange in biological sample preparation” (PONE-D-25-48079):

Comment: The title is concise, descriptive, and captures the core innovation of the work “low-cost,” “DIY,” and “programmable” are strong keywords that attract interest in open-source hardware and biological sample preparation.

Suggestions:

• Consider specifying

---

## [Decision Letter · Decision Letter 1]

2 Jan 2026

Design and evaluation of a low-cost, DIY programmable tissue processor for solvent exchange in biological sample preparation

PONE-D-25-48079R1

Dear Dr. Bacheva and co-authors,

We’re pleased to inform you that your manuscript has been judged scientifically suitable for publication and will be formally accepted for publication once it meets all outstanding technical requirements.

Kind regards,

Fentahun Adane Nigat, MSc., PhD

Academic Editor

PLOS One

Additional Editor Comments (optional):

Reviewers' comments:

Reviewer's Responses to Questions

**Comments to the Author**

Reviewer #1: All comments have been addressed

Reviewer #2: All comments have been addressed

2. Is the manuscript technically sound, and do the data support the conclusions?

Reviewer #1: Yes

Reviewer #2: Yes

3. Has the statistical analysis been performed appropriately and rigorously?

Reviewer #1: Yes

Reviewer #2: Yes

4. Have the authors made all data underlying the findings in their manuscript fully available?

Reviewer #1: Yes

Reviewer #2: Yes

5. Is the manuscript presented in an intelligible fashion and written in standard English?

Reviewer #1: Yes

Reviewer #2: Yes

Reviewer #1: I thank the authors for their rigorous revision. My comments were fully addressed. I think the revised manuscript was ready for publication.

Reviewer #2: I would like to thank the authors for their commitment to address all of the comments I have given to them.

**Do you want your identity to be public for this peer review?** For information about this choice, including consent withdrawal, please see our Privacy Policy

Reviewer #1: No

Reviewer #2: **Yes:** Dr. Hussen Abdu Muhidin

---

## [Editor Report · Acceptance letter]

PONE-D-25-48079R1

PLOS One

Dear Dr. Bacheva,

I'm pleased to inform you that your manuscript has been deemed suitable for publication in PLOS One. Congratulations! Your manuscript is now being handed over to our production team.

Kind regards,

on behalf of

Dr. Fentahun Adane Nigat

Academic Editor

PLOS One